# Insertion of a Clinical Pathway Pop-Up Window into a Computer-Based Prescription System: A Method to Promote Antibiotic Stewardship in Upper Respiratory Tract Infection

**DOI:** 10.3390/antibiotics10121479

**Published:** 2021-12-02

**Authors:** Wantin Sribenjalux, Nattawat Larbsida, Sittichai Khamsai, Benjaphol Panyapornsakul, Phitphiboon Deawtrakulchai, Atibordee Meesing

**Affiliations:** 1Department of Medicine, Faculty of Medicine, Khon Kaen University, Khon Kaen 40002, Thailand; nattawat@kkumail.com (N.L.); sittikh@kku.ac.th (S.K.); benjaphol.p@kkumail.com (B.P.); tumpery@hotmail.com (P.D.); atibordee@hotmail.com (A.M.); 2Division of Infectious Diseases and Tropical Medicine, Department of Medicine, Faculty of Medicine, Khon Kaen University, Khon Kaen 40002, Thailand; 3Research and Diagnostic Center for Emerging Infectious Diseases (RCEID), Khon Kaen University, Khon Kaen 40002, Thailand; 4Division of Ambulatory Medicine, Department of Medicine, Faculty of Medicine, Khon Kaen University, Khon Kaen 40002, Thailand

**Keywords:** antibiotic stewardship, antibiotic prescription, clinical pathway, pop-up window, upper respiratory tract infection

## Abstract

Outpatient antibiotics are most frequently prescribed for upper respiratory tract infection (URI); however, most such prescriptions are inappropriate. We aimed to determine the effect of an electronic clinical pathway on the rates of overall and rational prescription of antibiotics in patients with URI. A pilot quasi-experimental study was conducted in a university hospital and two of its nearby primary care units (PCU) in northeast Thailand from June to September 2020. Clinical pathway pop-up windows were inserted into the hospital’s computer-based prescription system. Care providers were required to check the appropriate boxes before they were able to prescribe amoxicillin or co-amoxiclav. We examined a total of 675 visits to the outpatient department due to URI at three points in time: pre-intervention, immediately post-intervention, and 6 weeks post-intervention. Patients in the latter group tended to be younger and visits were more likely to be general practitioner-related and to the student PCU than in the other two groups. In addition, the rate of antibiotic prescription was significantly lower at 6 weeks after intervention than at either of the other time periods (32.0% vs 53.8% pre-intervention and 46.2% immediately post-intervention; *p* < 0.001), and the proportion of rational antibiotic prescriptions increased significantly after implementation. Antibiotic prescription rates were lower at the community primary care unit and higher when the physician was a resident or a family doctor. The deployment of an electronic clinical pathway reduced the rate of unnecessary antibiotic prescriptions. The effect was greater at 6 weeks post-implementation. However, discrepancy of patients’ baseline characteristics may have skewed the findings.

## 1. Introduction

The spread of multidrug-resistant (MDR) bacteria is one of the most urgent global crises we currently face. In the United States, MDR bacteria is associated with approximately 100,000 in-hospital deaths annually and results in an economic burden of about USD 8 billion [1]. In Thailand, there are more than 100,000 cases of infection caused by MDR bacteria annually, resulting in approximately 38,000 deaths and costing around THB 40 billion (0.6 percent of Thailand’s gross domestic product (GDP)) [2]. One reason for the rise in drug-resistant strains is the widespread use of antibiotics, which are often inappropriately prescribed [3].

In primary care settings, antibiotics are most frequently used to treat upper respiratory tract infection (URI). Such cases account for 75% of all antibiotic use in Ireland and 77% in Thailand [4,5]. Although antibiotics are available over-the-counter in pharmacies in Thailand [6], public and private health care facilities still provide the majority (70.3%) of such medications [7]. The pre-intervention prescription rate was approximately 45% in patients with URI (despite the national target being under 20%) [8], and the most frequently prescribed antibiotics were amoxicillin and co-amoxiclav [9]. A recent study showed inappropriate antibiotic prescription to be common in URI [10]. Moreover, upper respiratory tract symptoms can also derive from atopic conditions (e.g., allergic rhinitis) [11], which may lead to unnecessary antibiotic prescription if care providers do not keep this in mind.

The use of a clinical pathway has been shown to reduce antibiotic use in URI [12,13,14]. However, the method of implementation may depend on how medication is dispensed. In hospitals where medication can be prescribed electronically, the insertion of clinical pathway pop-up windows is a convenient way to help physicians reconsider the necessity of antibiotics before prescription. This study was conducted as part of an antibiotic stewardship program (ASP) to determine the effectiveness of clinical pathway pop-up windows in URI by comparing the rates of antibiotic prescription before and after implementation. We also aimed to determine the factors associated with overall and rational antibiotic prescription in URI.

## 2. Results

### 2.1. Baseline Characteristics of Patients

Data from 675 outpatient department (OPD) visits due to URI were collected at three points during the study period: pre-intervention, immediately post-intervention, and 6 weeks post-intervention. The clinical pathway pop-up windows were added to the hospital’s electronic prescription process on 15 July 2020. Baseline data were collected for patients, including age, sex, medical conditions, symptoms, duration of illness, and recurrent URI history, as well as for the examining physician, including age, specialization, and location (Table 1). The patients’ mean age was 34.9 years, 31.7% were male, and most had no medical conditions. Those included in the pre-intervention and immediate post-intervention groups had similar baseline characteristics, with the exception that exudative tonsillitis and tenderness of cervical lymph nodes were more common in the latter group. Patients in the 6 weeks post-intervention group were more likely to be younger, present with cough, have no recurrent URI history, and present at the student primary care unit (PCU), and were less likely to present at a premium clinic or ear nose throat (ENT) OPD. The physicians who provided care for patients in this group were also younger and more likely to be general practitioners (GP) than those in the other groups.

### 2.2. Antibiotic Prescription Rate and Its Rationale

The rate of antibiotic prescription was significantly lower 6 weeks after intervention than at the other time periods (32.0% vs. 53.8% pre-intervention and 46.2% immediately post-intervention; *p* < 0.001; Table 2). In addition, the rate of rational antibiotic prescription increased with duration after intervention (79.4% at 6 weeks, 77.5% at immediately after intervention, and 60.6% pre-intervention; *p* < 0.001). However, when using cases with antibiotic prescription as a denominator, the rate of rational antibiotic prescription was highest immediately post-intervention (52.8%), differing significantly from those pre-intervention (21.3%) and 6 weeks after intervention (38.2%; *p* < 0.001). Mean duration of prescribed antibiotic treatment was similar in all three groups. After implementation of the clinical pathway, prescriptions shifted toward amoxicillin and away from co-amoxiclav. In the pre-intervention group, co-amoxiclav accounted for 48.8% of antibiotic prescriptions and amoxicillin only 40.5%. However, immediately post-intervention and at 6 weeks, the prescription rate of amoxiclav dropped to 34.0% and 23.6%, respectively, and that of amoxicillin increased to 54.4% and 68.0%.

### 2.3. Factors Associated with Antibiotic Prescription

According to univariate analysis, clinical presentations significantly associated with antibiotic prescription were exudative tonsillitis, cervical lymph node tenderness, purulent nasal discharge, and facial tenderness, with odds ratios of 188.5, 16.5, 3.71, and 3.34, respectively (Table 3). Patients who presented with cough were less likely to receive antibiotics, with an odds ratio of 0.36. Fever was not associated with physicians’ decision to prescribe antibiotics. Use of systemic corticosteroids at any dose was the only factor related to medical history associated with antibiotic prescription, with an odds ratio of 9.31. In terms of physician-related factors, those over 30 years of age, with specializations or sub-specializations other than internal medicine, and working at premium clinics, tended to prescribe more antibiotics than younger physicians, general practitioners, and those working at general OPD with odds ratios of 2.00, 3.06, and 3.11, respectively. Implementation of the clinical pathway (at both time periods) and being at a community PCU were associated with fewer antibiotic prescriptions (crude odds ratios (cOR): 0.55 and 0.35).

Statistically significant variables according to univariate analysis were included in subsequent multivariate analysis. After adjustment for physician’s age, specialization, and practice setting, the clinical pathway was still significantly associated with fewer antibiotic prescriptions with an adjusted odds ratio of 0.62 (95% confidence interval 0.43–0.89, *p* = 0.010; Table 4). The rate of antibiotic prescription was lower in community PCUs (aOR = 0.16, 95% CI of 0.05–0.48, *p* = 0.001) and higher when the physician was a resident or family doctor, with adjusted odds ratios of 1.66 (95% CI of 1.07–2.56, *p* = 0.022) and 10.05 (95% CI of 1.97–51.36, *p* = 0.006), respectively.

### 2.4. Factors Associated with Rational Antibiotic Prescription

Seven factors were associated with rational antibiotic prescription according to univariate analysis. Cough was the only clinical presentation related with fewer rational antibiotic prescriptions. Physicians over 30 years of age, with specializations or sub-specializations other than internal medicine, who were family doctors, and who worked at premium clinics, were more likely to prescribe antibiotics without adequate evidence of bacterial infection. Clinical pathway use and being at a community PCU were factors associated with higher rates of rational antibiotic prescription. These factors were subjected to multivariate analysis, according to which, use of the clinical pathway and being in a community PCU setting were associated with higher rates of rational antibiotic prescription, with adjusted odds ratios of 2.44 (95% CI of 1.61–3.70, *p* < 0.001) and 10.50 (95% CI of 1.96–56.27, *p* = 0.006), respectively. The physician being a resident was associated with lower rates (see Appendix A).

## 3. Discussion

### 3.1. Clinical Pathway as a Part of an Antibiotic Stewardship Program

This was a pilot quasi-experimental study, which demonstrated the effectiveness of implementing a clinical pathway in the hospital’s computer system in accordance with antibiotic stewardship measures. Our results showed that this intervention reduced the rate of overall antibiotic prescription and increased that of rational prescription in patients with URI after adjustment for the physician’s age, practice setting, and specialization. It also resulted in a shift away from broad-spectrum to narrow-spectrum antibiotics (co-amoxiclav to amoxicillin). These results were similar to those of Jenkin et al., who reported an 11.2% relative reduction in antibiotic prescription for non-pneumonia respiratory infection and a 14.4% relative reduction in broad-spectrum antibiotic use during the intervention period [12]. In addition, Madran et al. reported that the rate of antibiotic prescription decreased from 49% pre-implementation of an antibiotic stewardship program (ASP) to 29% post-implementation and that the application of a clinical pathway resulted in a rate of appropriate antibiotic use of 82% [13]. However, our application of the clinical pathway differed from those in the studies above in that, instead of providing a hard copy, we inserted clinical pathway pop-up windows into the electronic prescription system. This allowed the intervention to be more relevant in a hospital where prescriptions are managed electronically while still adhering to the core concept of antibiotic stewardship. Physicians were able to review the indications for antibiotic administration using the checkboxes provided and receive information regarding proper management of certain cases. The lower prescription rate after six weeks suggested that the educational component of the intervention had been effective. Previous studies have similarly found that physicians required time to learn and improve their practice [15,16,17,18].

### 3.2. Factors Associated with Antibiotic Prescription

We found that patients with exudative tonsillitis, cervical lymph node tenderness, purulent nasal discharge, or facial tenderness, on physical examination, had higher odds ratios of antibiotic prescription than those without, comparable to the findings of a previous study [19]. Fever, which may occur either due to bacterial or viral infection was not a predictor of antibiotic use, and cough, which is more suggestive of viral than bacterial infection in acute pharyngitis [20,21], was associated with a lower rate of antibiotic prescription in this study. This suggests that physicians were aware of the differing presentation of each etiology and were likely to comply with the treatment guidelines. A review from Ireland found that mid- or late-career physicians were more likely to prescribe antibiotics [4], but another study in the US reported that the antibiotic prescription rate was inversely correlated with age [22]. We also found that physicians older than 30 years were more likely to prescribe antibiotics according to the univariate but not the multivariate model. Some previous studies have also reported that specialists tend to prescribe antibiotics at a lower rate than GPs, but others have found no association [22,23]. We found that the physician being a GP was independently associated with a lower rate of antibiotic prescription when compared to residents and family doctors. These differences among various studies are likely due to differences in national health care systems, culture, and the level of the hospital in which a given study was conducted. GPs in Srinagarind Hospital are mostly young physicians who graduated during the national rational drug use (RDU) campaign [8]. However, residents in fields other than internal medicine may be more focused on improving their specialized skills and knowledge than on general topics. Family doctors, most of whom are senior preclinical staff with medical licenses and degrees in family medicine, tended to prescribe antibiotics for most patients with URI. This may be due to the demands of working in a high-volume environment not affording them the opportunity to make certain diagnoses [23]. However, many of these physicians also have positive attitudes regarding antibiotic use and believe antibiotic administration to be the best course of treatment for their patients [24]. These findings may assist hospitals in creating effective policies to reduce unnecessary antibiotic prescription, which can lead to adverse events and the proliferation of drug-resistant microorganisms [25,26].

### 3.3. Factors Associated with Rational Antibiotic Prescription

In this study, factors associated with rational antibiotic use were quite similar to those associated with overall antibiotic use mentioned above. Clinical pathway use improved the rate of rational antibiotic prescription both immediately post-intervention and 6 weeks after. This may have been because the interactive pop-up windows provided the researchers with more clinical data to judge whether antibiotic prescriptions were rational or not. One factor associated with irrational prescription was greater physician age. This is similar to previous findings. Aspinall et al., for example, reported that providers ≥30 years of age prescribed more antibiotics for nonbacterial acute respiratory tract infections with an odds ratio of 2.6 and 95% confidence interval of 1.1 to 6.3 [23]. Zoutman et al. also found that physician age was associated with not prescribing first-line antibiotics for streptococcal pharyngitis with an odds ratio of 1.6 and 95% confidence interval of 1.3 to 2.1 [24]. This may be due to the generational effect, practice styles, and awareness of antibiotic overuse [23].

### 3.4. *Limitations*

There were some limitations to this study. First, baseline characteristics in the three groups differed. This was especially true at 6 weeks post-intervention. Both patients and physicians in this group tended to be younger and visits were more likely to be GP-related and to the student PCU. Because the study site is a university hospital, we had planned to enroll all patients after the beginning of the semester in order to include more students. However, a series of COVID-19 outbreaks led to the new semester being postponed. This meant that the patients seen during the pre-intervention and immediate post-intervention periods were mostly university personnel, who were older and more likely to visit a premium clinic. This may have skewed the findings, as the setting being a premium clinic was associated with antibiotic prescription in the univariate but not multivariate model. Patients seeking antibiotics can also affect physicians’ decisions [24], but this study did not compare attitudes toward antibiotic use between university personnel and students. Further study may be required to prove the true effect of this intervention while controlling for these factors. Second, despite the prospective study design, there were some missing data, especially regarding clinical presentation. As URI is normally an uncomplicated disease, physicians rarely take the time to record all symptoms in patients’ medical records. Third, the social and cultural context likely affects the use and misuse of antibiotics. Thus, the factors we found to be significant may not be so in other regions/contexts. Finally, the insertion of clinical pathway checklist pop-up windows into hospital computer systems may not be applicable in all settings. The method of intervention must be adjusted to best suit the prescription process of the hospital at which it is being applied. Despite these limitations, this study provides valuable evidence that implementing a clinical pathway in the form of interactive pop-up windows may help reduce the rate of antibiotic prescriptions and increase the rate of rational antibiotic use. This method can be implemented along with other interventions as a part of ASPs to help ensure the optimal use of antibiotics.

## 4. Materials and Methods

### 4.1. Study Design and Setting

This was a pilot quasi-experimental study conducted in a 1400-bed university hospital and two of its nearby PCUs in northeast Thailand: PCU KKU123 (student PCU) and PCU Samliam (community PCU), from June to September 2020. We inserted clinical pathway pop-up windows (available in the Appendix A) into the hospital’s electronic prescription system. When physicians attempted to prescribe amoxicillin or co-amoxiclav, these pop-up windows would appear, warning that “90 to 98% of rhinosinusitis cases are caused by viral infection, for which antibiotic treatment is not necessary” and that “only 5–8% of acute pharyngitis cases are caused by group A streptococci and require antibiotic treatment to prevent rheumatic heart disease” [27,28]. Before the prescription was processed, physicians were required to mark checkboxes specifying the indications of the antibiotic. Regardless of their response, they would still have the option to complete the prescription. They could also skip the pop-up windows if they indicated that the antibiotics were meant to treat conditions other than URI. Clinical data were collected for patients over 18 years of age with a principal diagnosis of URI during the study periods. We excluded the patients with documented beta-lactam allergy and whose condition was initially misidentified according to their ICD-10 Clinical Modification (ICD-10 CM) code. The primary outcome of the study was the reduction rate of overall antibiotic prescription after implementing intervention. Rates of rational prescription before and after intervention were also compared. Patients’ electronic medical records were thoroughly reviewed. Data collected included baseline characteristics, clinical presentation, setting, physician’s age and specialty, and duration and type of prescribed antibiotics. These factors were analyzed for any association with antibiotic prescription and rational prescription as secondary outcomes.

### 4.2. Operational Definitions

#### 4.2.1. Upper Respiratory Tract Infection (URI)

Defined by an ICD-10 CM code of J00 to J03 as principal diagnosis or co-morbidity assigned by the patient’s care provider at their visit to the OPD.

#### 4.2.2. Rational Antibiotic Prescription

For patients with ICD-10 CM codes J00 or J01, antibiotic prescription was considered rational if clinical presentation met at least one of the following criteria [27]:Symptoms or signs compatible with acute rhinosinusitis, lasting for at least 10 days without clinical improvement;Severe symptoms (defined as fever ≥39 °C and purulent nasal discharge or facial pain) lasting for at least 3–4 consecutive days at the beginning of illness;Worsening of symptoms or signs (characterized by the new onset of fever, headache, or increase in nasal discharge) following a typical viral URI that lasted 5–6 days and that were initially improving.

For patients with ICD-10 CM codes J02 or J03, antibiotic prescription was considered rational if the patient had a cumulative modified Centor score ≥2 [29].

#### 4.2.3. Premium Clinic

Name of clinic at Srinagarind Hospital which opens after hours and provides care only by attendance. There are doctor’s fees and additional service charges in this clinic.

### 4.3. Sample Size Calculation

Sample size calculation was based on the primary objective of the study. Based on a pilot study, the rate of antibiotic prescription in URI patients at Srinagarind Hospital prior to implementation was 32%. Assuming that the intervention would reduce the prescription rate to the national goal of below 20% [5], we determined that a sample size of at least 209 participants per group would be necessary. In spite of the higher rate of prescription in the study period, the study still had a power of more than 90% to differentiate between antibiotic prescription rates before and after intervention.

### 4.4. Statistical Analysis

Categorical variables were reported as number and percentage. Continuous variables were presented as mean with standard deviation (SD). Pearson’s chi-squared test was performed to compare the percentage of cases with the rates of overall and rational antibiotic prescription at pre-intervention, immediately post-intervention, and 6 weeks post-intervention. Fisher’s exact test was performed in cases where chi-square was not suitable. One-way analysis of variance (ANOVA) was used to compare the mean of each group. Associations were expressed as odds ratio (OR) and 95% confidence interval (95% CI). Factors possibly associated with antibiotic prescription and rational use were subjected to univariate analysis, and those that were statistically significant (*p*-value < 0.05) were selected for multivariate logistic regression. All statistical analyses were performed using SPSS for Windows, version 26.0 (SPSS).

### 4.5. Ethical Consideration

Ethical approval was provided by the Center for Ethics in Human Research, Khon Kaen University, in accordance with the Declaration of Helsinki (Number HE621517).

## 5. Conclusions

Insertion of clinical pathway pop-up windows into a hospital computer-based prescription system reduced unnecessary antibiotic prescription and led to a shift from broad-spectrum to narrow-spectrum antibiotics. This effect was greater at 6 weeks after intervention. We also found that residents and family doctors were more likely to prescribe antibiotics than GPs, and that patients visiting a community PCU were less likely to receive antibiotics. Hospital administrators should be conscious of these factors and consider implementing the method described here in combination with other effective interventions in order to reduce antibiotic misuse and improve the quality of care.

## Figures and Tables

**Table 1 antibiotics-10-01479-t001:** Baseline characteristics of patients in the three phases of the study.

Baseline Characteristics	Total (N = 675)	Pre-Intervention (N = 225)	Immediately Post-Intervention (N = 225)	6 Weeks after Intervention (N = 225)	*p*-Value *
Data collection period		2 June–15 July, 2020	15 July–8 August, 2020	1–14 September, 2020	NA
Age (mean ± SD)	34.9 (16.7)	38.7 (17.1)	35.7 (17.2)	30.2 (14.7)	<0.001
Sex (male, %)	214 (31.7)	76 (35.5)	76 (35.5)	62 (29.0)	0.260
Medical conditions (N, %)					
• DM	24 (3.6)	7 (3.1)	12 (5.3)	5 (3.2)	0.185
• HT	31 (4.6)	17 (7.6)	6 (2.7)	8 (3.6)	0.031
• Airway disease	14 (2.1)	8 (3.6)	3 (1.3)	3 (1.3)	0.199
• Steroid use	8 (1.2)	1 (0.4)	4 (1.8)	3 (1.3)	0.426
Clinical presentation (N, %) **					
• Fever	250 (39.6)	64 (29.8)	92 (46.2)	94 (43.1)	0.001
• Cough	235 (62.0)	54 (56.3)	70 (53.0)	111 (75.3)	0.001
• Exudate on tonsil	82 (29.8)	16 (20.0)	41 (53.9)	25 (21.0)	<0.001
• CLN tenderness	44 (35.2)	7 (15.9)	18 (44.2)	19 (17.4)	0.004
• Purulent nasal discharge	42 (24.3)	17 (21.0)	19 (23.5)	6 (54.5)	0.050
• Facial tenderness	34 (22.8)	13 (33.3)	5 (31.3)	16 (17.0)	0.087
History of recurrent URI within 6 months	134 (19.9)	56 (24.9)	49 (21.8)	29 (12.9)	0.004
Duration of illness (days, mean ± SD)	3.20 ± 3.08	3.43 ± 3.74	3.08 ± 2.71	3.13 ± 2.78	0.535
Setting (N, %)					
• Student PCU	238 (35.2)	50 (22.2)	79 (35.1)	109 (48.4)	<0.001
• ED	155 (23.0)	53 (23.6)	46 (20.4)	56 (24.9)	0.271
• Premium clinic	82 (12.1)	32 (14.2)	34 (15.1)	16 (7.1)	0.005
• General OPD	60 (8.9)	20 (8.9)	22 (9.8)	18 (8.0)	0.566
• ENT OPD	51 (7.6)	28 (12.4)	15 (6.7)	8 (3.6)	<0.001
• Community PCU	44 (6.5)	22 (9.8)	13 (5.8)	9 (4.0)	0.015
• Medicine OPD	10 (1.5)	4 (1.8)	3 (1.3)	3 (1.3)	0.652
• Others	35 (5.2)	16 (7.1)	13 (5.8)	6 (2.7)	0.036
Type of physician					
• GP	307 (45.5)	54 (24.0)	106 (47.1)	147 (65.3)	<0.001
• Resident	200 (29.7)	91 (40.5)	62 (27.5)	47 (20.9)	<0.001
• Other specialist ^$^	129 (19.1)	59 (26.2)	49 (21.8)	21 (9.4)	<0.001
• Internist	23 (3.4)	14 (6.2)	2 (0.9)	7 (3.1)	0.004
• Family doctor	13 (1.9)	7 (3.1)	4 (1.8)	2 (0.9)	0.113
• ID physician	3 (0.4)	0 (0.0)	2 (0.9)	1 (0.4)	0.220
Physician’s age (mean ± SD)	30.9 (10.2)	33.3 (12.0)	30.7 (10.0)	28.6 (7.6)	<0.001

CLN: cervical lymph node, DM: diabetes mellitus, ED: emergency department, ENT: ear nose throat, GP: general practitioner, HT: hypertension, ID: infectious disease, OPD: outpatient department, PCU: primary care unit, URI: upper respiratory infection; * *p* < 0.05—significant difference in at least one of the three groups; ** Excluded missing data in denominators; ^$^ Other specialist included otolaryngologist, radiologist, rehabilitation physician, and psychiatrist.

**Table 2 antibiotics-10-01479-t002:** Antibiotic prescription and its rationale at baseline and intervention periods.

Outcomes (All Cases)	Baseline Period (N = 225)	Immediately Post-Intervention (N = 225)	6 Weeks after Intervention (N = 225)	*p*-Value *
• Number of ATB prescriptions (%, 95% CI)	53.8 (47.2–60.3)	46.2 (39.7–52.8)	32.0 (25.9–38.1)	<0.001
• Rational ATB prescriptions (%, 95% CI)	60.6 (53.6–67.7)	77.5 (71.8–83.1)	79.4 (74.0–84.9)	<0.001
Outcome (ATB prescription cases)	Baseline period (N = 121)	Immediate intervention period (N = 104)	6 weeks after intervention (N =7 2)	*p*-value *
• Date of therapy (mean ± SD, days)	9.4 ± 2.7	9.2 ± 2.7	9.1 ± 2.9	0.643
• Rational ATB prescriptions (%, 95% CI)	21.3 (12.9–29.7)	52.5 (42.5–62.5)	38.2 (26.4–50.1)	<0.001
• Type of ATB (N, %)				
○ Amoxicillin	49 (40.5)	56 (54.4)	49 (68.0)	<0.001
○ Co-amoxiclav	59 (48.8)	35 (34.0)	17 (23.6)	<0.001
○ Macrolides	4 (3.3)	6 (5.8)	3 (4.2)	0.363
○ Quinolones	6 (4.9)	3 (2.9)	3 (4.2)	0.492
○ Others	3 (2.5)	3 (2.9)	0 (0.0)	0.238

ATB: antibiotic, CI: confidence interval; * *p* < 0.05—significant difference in at least one of the three groups.

**Table 3 antibiotics-10-01479-t003:** Factors associated with antibiotic prescription.

Factor	cOR (95% CI)	***p***-Value
Sex (male)	1.21 (0.87–1.67)	0.255
DM	1.08 (0.48–2.45)	0.854
HT	1.81 (0.87–3.76)	0.111
Airway disease	0.95 (0.33–2.78)	0.931
Steroid use *	9.13 (1.12–74.64)	0.039
Fever	1.37 (1.00–1.89)	0.053
Cough *	0.36 (0.23–0.55)	<0.001
CLN tenderness *	16.5 (5.82–46.76)	<0.001
Exudate on tonsil(s) *	188.5 (28.62–1, 387.38)	<0.001
Facial tenderness *	3.34 (1.55–7.20)	0.002
Purulent nasal discharge *	3.71 (1.67–8.21)	0.001
History of recurrent URI within 6 months	1.35 (0.93–1.98)	0.119
Age ≥ 40	0.88 (0.63–1.22)	0.448
Physician’s age ≥ 30 *	2.00 (1.43–2.79)	<0.001
Setting (reference = general OPD)		
• Student PCU	1.17 (0.65–2.11)	0.602
• ED	1.79 (0.96–3.31)	0.065
• Premium clinic *	3.06 (1.53–6.11)	0.002
• ENT OPD	2.08 (0.97–4.49)	0.059
• Community PCU *	0.35 (0.13–0.92)	0.034
• Medicine OPD	0.46 (0.09–2.38)	0.359
Type of physician (reference = GP)		
• Resident *	1.45 (1.01–2.08)	0.047
• Specialization other than internal medicine ^$,^ *	3.11 (2.03–4.77)	<0.001
• Internist	1.42 (0.60–3.34)	0.425
• Family doctor *	6.14 (1.66–22.79)	0.007
• ID physician **	NA	NA
**Clinical pathway use ***	**0.55 (0.40–0.76)**	**<0.001**

CLN: cervical lymph node, CI: confidence interval, cOR: crude odds ratio, DM: diabetes mellitus, ED: emergency department, ENT: ear nose throat, GP: general practitioner, HT: hypertension, ID: infectious disease, NA: not available, OPD: outpatient department, PCU: primary care unit, URI: upper respiratory infection; * Factor with statistical significance; ** Number too low to calculate; ^$^ “Specialization other than internal medicine” included otolaryngologist, radiologist, rehabilitation physician, and psychiatrist.

**Table 4 antibiotics-10-01479-t004:** Factors associated with antibiotic prescription (multivariate analysis) when adjusted for physician’s age, location, specialization, and clinical pathway usage.

Factor	aOR (95% CI)	*p*-Value
Physician’s age ≥ 30	1.12 (0.52–2.40)	0.779
**Clinical pathway usage ***	**0.62 (0.43–0.89)**	**0.010**
Type of physician (Reference = GP)		
• Resident *	1.66 (1.07–2.56)	0.022
• Specialization other than internal medicine ^$^	1.99 (0.75–5.30)	0.168
• Family doctor *	10.05 (1.97–51.36)	0.006
Location (Reference = GP OPD)		
• Premium clinic	1.65 (0.63–4.34)	0.307
• Community PCU *	0.16 (0.05–0.48)	0.001

aOR: adjusted odds ratio, CI: confidence interval, GP: general practitioner, OPD: outpatient department, PCU: primary care unit; * Factor with statistical significance; ^$^ “Specialization other than internal medicine” included otolaryngologist, radiologist, rehabilitation physician, and psychiatrist.

## Data Availability

Data are available on reasonable request and in line with the permission approval processes from the Faculty of Medicine, Khon Kaen University, Khon Kaen, Thailand.

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
