# Peer review of "Insertion of a Clinical Pathway Pop-Up Window into a Computer-Based Prescription System: A Method to Promote Antibiotic Stewardship in Upper Respiratory Tract Infection"

_antibiotics, 2021, doi:10.3390/antibiotics10121479_

Round 1

Reviewer 1 Report

This is a very useful paper showing how computer pop-ups can be used to improve antibiotic prescribing.

I have two main concerns:

1) I'm unclear why the methods are presented after the discussion (unless this is a  journal 'style' policy?). It is most unusual.  I think this section (4 in the manuscript) should come before the Results (Section 2 in the manuscript). This would make the research much easier to follow.

2) The limitations section (3.4) briefly describes why the characteristics of the people treated, and where they were treated, 6 weeks post intervention differed so markedly from those pre-intervention and immediately post-intervention. I initially noted these differences in Table 1 and thought the discrepancy was odd. You advise that this was because COVID-19 stopped students returning to the university hospital until this later period, and they were younger and students. I can understand how this upset the intentions of the study, but it is a major limitation as it affects how antibiotics may be used for URI. I think this problem needs to be stated clearly in the abstract and discussed in more detail. It could obviously skew the findings, and unfortunately further work may be required to understand this further.

Author Response

Reviewer 1

This is a very useful paper showing how computer pop-ups can be used to improve antibiotic prescribing.

I have two main concerns:

  • I'm unclear why the methods are presented after the discussion (unless this is a journal 'style' policy?). It is most unusual.  I think this section (4 in the manuscript) should come before the Results (Section 2 in the manuscript). This would make the research much easier to follow.

Ans: Thank you for your kind suggestion. As an author, I also agree with your opinion, but this is a journal 'style' policy.

  • The limitations section (3.4) briefly describes why the characteristics of the people treated, and where they were treated, 6 weeks post intervention differed so markedly from those pre-intervention and immediately post-intervention. I initially noted these differences in Table 1 and thought the discrepancy was odd. You advise that this was because COVID-19 stopped students returning to the university hospital until this later period, and they were younger and students. I can understand how this upset the intentions of the study, but it is a major limitation as it affects how antibiotics may be used for URI. I think this problem needs to be stated clearly in the abstract and discussed in more detail. It could obviously skew the findings, and unfortunately further work may be required to understand this further.

Ans: Thank you for understanding this limitation. We stated this problem into our abstract part and gave more detailed in discussion part as your suggestion.

Abstract: “We examined a total of 675 visits to the outpatient department due to URI at three points in time: pre-intervention, immediately post intervention, and 6 weeks post intervention. Patients in the latter group tended to be younger and visits were more likely to be GP-related and to the student PCU than in the other two groups. In addition, the rate of antibiotic prescription was significantly lower at 6 weeks after intervention than at either of the other time periods (32.0% vs 53.8% pre-intervention and 46.2% immediately post intervention; P <0.001),”

Discussion: "This meant that the patients seen during the pre-intervention and immediate post-intervention periods were mostly university personnel, who were older and more likely to visit a premium clinic. This may have skewed the findings, as the setting being a premium clinic was associated with antibiotic prescription in the univariate but not multivariate model. Patients seeking antibiotics can also affect physicians’ decisions, (Zoutman D, et al. Int j infect control. 2008;4(1)), but this study did not compare attitudes toward antibiotic use between university personnel and students. Further study may be required to prove the true effect of this intervention while controlling for these factors. Second..."

Reviewer 2 Report

Good work about a topic often little attenzioned. Some english mistakes must be corrected. Only one precisely must be underlined: you should mention in the text (since the study is ended and therefore it can not added in the pop-up) the possibility that URTI beyond viral origin can derive from atopic condition (e.g. allergic rhinitis). About this, you must mention in your references:

- Martines F, Dispenza F, Sireci F, Gallina S, Salvago P. Eustachian Tube Function Assessment after Radiofrequency Turbinate Reduction in Atopic and Non-Atopic Patients. Int J Environ Res Public Health. 2021 Jan 20;18(3):881. 

Author Response

Reviewer 2

Good work about a topic often little attenzioned. Some english mistakes must be corrected. Only one precisely must be underlined: you should mention in the text (since the study is ended and therefore it can not added in the pop-up) the possibility that URTI beyond viral origin can derive from atopic condition (e.g. allergic rhinitis). About this, you must mention in your references:

- Martines F, Dispenza F, Sireci F, Gallina S, Salvago P. Eustachian Tube Function Assessment after Radiofrequency Turbinate Reduction in Atopic and Non-Atopic Patients. Int J Environ Res Public Health. 2021 Jan 20;18(3):881. 

 Ans: We appreciated for your kind suggestion. We added this possibility in the introduction part. It is very useful viewpoint that we would consider in our further study.

A recent study showed inappropriate antibiotic prescription to be common in URI [10]. Moreover, upper respiratory tract symptoms can also derive form atopic conditions (e.g., allergic rhinitis) (Martines F, et al. Int J Environ Res Public Health. 2021;18(3):881.), which may lead to unnecessary antibiotic prescription if care providers do not keep this in mind.”

Reviewer 3 Report

The study aims to determine the effect of an electronic clinical pathway on improving rational use of antibiotics in Upper respiratory tract infection. The study has been well written but few areas needs more explanation. 1. Introduction needs more literature evidence in the field and the context of Thailand 2. This could be explained somewhere. 3. Why only a university hospital was considered for sampling? Need more explanation 4. Sampling is unclear. In the abstract, author says only a university hospital, however, in the method, it says 3 sites. If three sites, is there any way to present effectiveness results by sites and by comparison 5. “A quasi-experimental study was conducted in a university hospital in northeast Thailand 19 from June to September 2020” (Abstract) “1400-bed university hospital and two of its nearby PCU in northeast Thailand (PCU KKU123 and PCU Samliam)” (Method) 6. What was the sample size for each site? 7. Factors associated with antibiotic prescription-was it a secondary outcome of the study? 8. The study outcomes can be explained in the method section. 9. Clinical pathway pop-up windows were specific for acute nasopharyngitis and acute sinusitis, acute tonsillitis and acute pharyngitis. How did author recruit patients for those categories? The clinical indications of table 3 are a bit confusing from the focus of those clinical indication of interest. 10. Inclusion and exclusion criteria of the patients included in the study could be explained. How did author recruit patients and/or consider prescribers? 11. Author mentions training before implementing intervention. This aspect could be explained in method. Target population, training modes. 12. Need more information about how pop up message interventions were incorporated and set up in study sites. 13. Why amoxicillin and co-amoxiclav were of interest?

Round 2

Reviewer 1 Report

I do not think you have addressed major concerns about this study and the presentation.

The patients seen at 6 weeks after intervention were a different population from those seen pre-intervention and immediately post-intervention. This was the result of COVID-19 - which might also have altered the way antibiotics were prescribed. This is unfortunate, but it means that not much can be drawn from the findings. At the least this should be discussed in the abstract.

The other problem I have with this paper is that the results and discussion are presented before the methods (which come last). This is highly unusual and makes the paper difficult to digest.

Author Response

Reviewer 1

I do not think you have addressed major concerns about this study and the presentation.

The patients seen at 6 weeks after intervention were a different population from those seen pre-intervention and immediately post-intervention. This was the result of COVID-19 - which might also have altered the way antibiotics were prescribed. This is unfortunate, but it means that not much can be drawn from the findings. At the least this should be discussed in the abstract.

Ans: Thank you for your kind suggestion. We had stated this problem into our abstract part by added the sentence “Patients in the latter group tended to be younger and visits were more likely to be GP-related and to the student PCU than in the other two groups.” into the paragraph. However, we additionally added “However, discrepancy of patients’ baseline characteristics may have skewed the findings.” as the last sentence to made it more crystal-clear.

The other problem I have with this paper is that the results and discussion are presented before the methods (which come last). This is highly unusual and makes the paper difficult to digest.

Ans: Thank you for your kind suggestion. As an author, I agree that presenting the results and discussion before the methods makes the paper difficult to digest, but this is a journal 'style' policy as mentioned before.

Reviewer 3 Report

This is a much improved version. However, because of a single centre study, the design of the study could be reported as a pilot quasi-experimental study. The relevant section might be revised with this term. 

Author Response

This is a much improved version. However, because of a single centre study, the design of the study could be reported as a pilot quasi-experimental study. The relevant section might be revised with this term. 

Ans: Thank you for your kind suggestion. We accept to change the word of study design to “a pilot quasi-experimental study” in all relevant parts.